# Loss Landscapes are All You Need: Neural Network Generalization Can Be Explained Without the Implicit Bias of Gradient Descent

**Ping-yeh Chiang[1], Renkun Ni[1], David Yu Miller[1,3], Arpit Bansal[1], Jonas Geiping[1], Micah Goldblum[2] & Tom Goldstein[1]**

[1]University of Maryland, College Park ,
{`pchiang,rn9zm,dym,bansal01,jgeiping,tomg`}`@umd.edu`
[2]New York University , `goldblum@nyu.edu`
[3]Max Planck Institute for Software Systems

## Abstract

It is commonly believed that the implicit regularization of optimizers is needed for neural networks to generalize in the overparameterized regime. In this paper, we observe experimentally that this implicit regularization behavior is *generic*, i.e. it does not depend strongly on the choice of optimizer. We demonstrate this by training neural networks using several gradient-free optimizers, which do not benefit from properties that are often attributed to gradient-based optimizers. This includes a guess-and-check optimizer that generates uniformly random parameter vectors until finding one that happens to achieve perfect train accuracy, and a zeroth-order Pattern Search optimizer that uses no gradient computations. In the low sample and few-shot regimes, where zeroth order optimizers are most computationally tractable, we find that these non-gradient optimizers achieve test accuracy comparable to SGD. The code to reproduce results can be found at `https://github.com/Ping-C/optimizer`.

## 1 Introduction

The impressive generalization of deep neural networks continues to defy prior wisdom, where over-parameterization relative to the number of data points is thought to hurt model performance. From the perspective of classical learning theory, using measures such as Rademacher complexity and VC dimension, as one increases the complexity of a model class, the generalization performance of learned models should eventually deteriorate. However, in the case of deep learning models, we observe the exact opposite phenomenon – as one increases the number of model parameters, the performance continues to improve. This is particularly surprising since deep neural networks were shown to easily fit random labels in the overparameterized regime (Zhang et al., 2017). This combination of empirical and theoretical pointers shows a large gap in our understanding of deep learning, which has sparked significant interest in studying various forms of implicit bias which could explain generalization phenomena.

Perhaps the most widely-held hypothesis posits that gradient-based optimization gives rise to implicit bias in the final learned parameters, leading to better generalization (Arora et al., 2019; Advani et al., 2020; Liu et al., 2020; Galanti & Poggio, 2022). For example, (Arora et al., 2019) showed that deep matrix factorization, which can be viewed as a highly simplified neural network, is biased towards solutions with low rank when trained with gradient flow. Indeed, (Galanti & Poggio, 2022) shows theoretically and empirically that stochastic gradient descent (SGD) with a small batch size can implicitly bias neural networks towards matrices of low rank. A related concept was used by (Liu et al., 2020) to show that gradient agreement between examples is indicative of generalization in the learned model.

In this paper, we empirically examine the hypothesis that gradient dynamics is a necessary source of implicit bias for neural networks. Our investigation is based on a comparison of several zeroth

order optimizers, which require no gradient computations, with the performance of SGD. We focus our studies on the small sample regime where zeroth order optimizations are tractable. Interestingly, we find that all the gradient-free optimizers we try generalize well compared to SGD in a variety of settings, including MNIST (LeCun et al., 2010), CIFAR-10 (Krizhevsky, 2009), and few-shot problems (Bertinetto et al., 2019; Vinyals et al., 2016).

Even though we use fewer samples in our experiments compared to standard settings, this low-data regime highlights the role of model bias, where the generalization behavior of neural networks is particularly intriguing. The model we test has more than $10,000$ parameters, but it has to generalize with fewer than $1,000$ training samples. Without implicit bias, such a feat is nearly impossible in realistic use cases like the ones we consider. Our work shows empirically that generalization does not require the implicit regularization of gradient dynamics, at least in the low-data regime. It is still an open question whether gradient dynamics play a larger role in other regimes, namely, where more data is available.

We need to caution that we are not claiming that gradient dynamics have no effect on generalization, as it has been clearly shown both theoretically and empirically that it has a regularizing effect (Arora et al., 2019; Galanti & Poggio, 2022). Instead, we argue that the implicit regularization of gradient dynamics is only secondary to the observed generalization performance of neural networks, at least in the low-data regimes we study.

The observations in this paper support the idea that implicit bias can come from properties of the loss landscape rather than the optimizer. In particular, they support the *volume hypothesis* for generalization: The implicit bias of neural networks may arise from the volume disparity of different basins in the loss landscape, with good hypothesis classes occupying larger volumes. The conjecture is empirically supported by the observation that even a "guess & check" algorithm, which randomly samples solutions from parameter space until one is found with low training error, can generalize well. The success of this optimizer strongly suggests that generalizing minima occupy a much larger volume than poorly generalizing minima in neural loss functions, and that this volume disparity alone is enough to explain generalization in the low-shot regime.

Finally, we show in a previously studied toy example that volume implicitly biases the learned function towards good minima, regardless of the choice of optimizer.

## 2 RELATED WORK

The capability of highly overparametrized neural networks to generalize remains a puzzling topic of theoretical investigations. Despite their high model complexity and lack of strong regularization, neural networks do not overfit to badly generalizing solutions. From a classical perspective, this is surprising. Bad global solutions do exist (Zhang et al., 2017; Huang et al., 2020b), yet usual training routines which optimize neural networks with stochastic gradient descent never find such worst-case solutions. This has led a flurry of work re-characterizing and investigating the source of the generalization ability of neural networks. In the following we highlight a few angles.

**High-dimensional optimization**   Before reviewing the literature on gradient dynamics, we want to review the underlying reasons why gradient-based (first-order) optimization is so central to deep neural networks: The core reasons for this is often dubbed the *curse of dimensionality*: For arbitrary optimization problems (with minimal conditions, i.e. (Noll, 2014)) a first-order optimizer will converge to a local minimal solution in polynomial time in the worst-case, independent of the dimensionality of the problem. However, a zeroth order algorithm without gradient information will have to, in the worst-case, evaluate a number of queries that increases exponentially with the dimensionality of the problem, even for smooth, convex optimization problems (Nesterov, 2004). However as we will discuss, neural networks are far from a worst-case scenario, given that many solutions exist due to the flatness of basins and the inter-connectedness of minima in neural networks.

**Gradient dynamics**   Here we briefly review literature that argues for gradient descent as the main implicit bias for generalization of neural networks. In Liu et al. (2020), they argue that deep networks generalize well because of the large agreement of gradients among training examples using a quantity called gradient signal-to-noise ratio (GSNR). They found both empirically and theoretically that a large GSNR would lead to better generalization and that deep networks induce a large GSNR during training, leading to better generalization. Arora et al. (2019) show that the dynamics of gradient-based optimization induce implicit bias that is stronger than typical norm-based bias in the setting of deep matrix factorization, and raise the question whether implicit biases can be induced from first-order optimization that cannot be captured by any explicit regularization. Advani et al.

(2020) argues that in the overparameterized regime, the gradient dynamics prevent learning from happening in a certain subspace of the weights, which effectively works as implicit regularization. A recent paper by (Galanti & Poggio, 2022) proves that SGD trained networks have a low-rank implicit bias and hypothesizes that such an implicit bias may be the source of superior generalization for deep neural networks.

**Non-gradient based explanation of implicit bias**  Several works have tried to explain the generalization behavior of neural networks with other forms of implicit regularization. Neyshabur et al. (2015) argues weight norms to be the main measure of capacity control that allows neural networks to generalize. Keskar et al. (2016) suggests that flatness in the parameter space corresponds to simpler functions, thus allowing neural networks to generalize. However, Dinh et al. (2017) later show that when the flatness measure is not scale-invariant, sharp solutions can generalize just as well with appropriate rescaling of the network parameters. Valle-Perez et al. (2018) argue that the parameter-function map is exponentially biased towards simple functions. Rahaman et al. (2019) shows that neural networks are biased toward low frequency functions that vary globally without local fluctuation. Among all works that try to explain neural network generalization, most recent works argue gradient descent or stochastic gradient descent as the main implicit bias of neural network training that allows deep overparameterized networks to generalize.

**Volume and Bayesian modeling**  From a Bayesian perspective, flat minima of the loss surface are highly represented in the Bayesian model average, especially when they contain functional diversity (Wilson & Izmailov, 2020). The size of a posterior peak has also been connected to Occam factors indicating that they represent simpler solutions which are more compressible and generalize well (MacKay et al., 2003). Smith & Le (2017) studies generalization behavior of overparameterized linear models where they find that the Bayesian evidence or marginal likelihood, which is connected to generalization, strongly favors flat minima. A line of work on PAC-Bayes generalization bounds, which is related to compressibility and the Bayesian evidence, uses compressibility to guarantee generalization and finds the flat minima are more compressible as they yield more bits back from the KL-divergence term in the bound (Dziugaite & Roy, 2017). In contrast to these works, our findings focus not on why flat minima generalize well but rather how their large volume makes them likely to be found by optimizers.

**Similar Lines of Inquiry**  Mingard et al. (2021) empirically show that when sampling from wide networks conditioned on the training set, the sampled models behave similarly to finite width networks trained with SGD. They approximate the posterior with Neural Network Gaussian Process, which is not exact in finite width networks. Geiping et al. (2022) show that full batch gradient descent, when coupled with explicit regularization can perform comparably to model trained with SGD, thus bringing into question the importance of SGD for generalization. Similar in spirit, we argue that SGD and all gradient-based optimizers are not the main source of generalization behavior of neural networks. Huang et al. (2020a) provide intuitive explanations for the volume hypothesis, and empirically measure the volume of both good and bad minima. While they show that individual good minima tend to have much larger volume than individual bad ones, their experiments do not show that the total volume of all good minima is large. Experiments below address this weakness.

## 3 THE MYSTERY OF GENERALIZATION WITH OVERPARAMETERIZATION

In this section, we illustrate how the complexity of a hypothesis class increases with the number of parameters in the context of a simple classification problem. Specifically, we increase the number of hidden units of a two layer neural network and showcase the increasing complexity of the model class. Despite this increased complexity, SGD often consistently finds good classifiers. Then, we proceed to show that a similar generalization behavior can be achieved without any gradient dynamics.

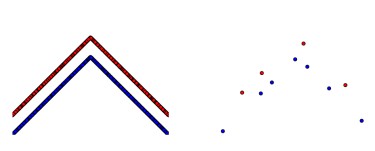

We begin with a toy classification problem defined over two classes where the data distribution is a wedge "Λ" with a vertical margin separating the two classes (see Figure 1). Throughout this section, our training and testing data consist of 11 points and 5 points, respectively, each sampled uniformly at random.

Figure 1: On the left, we have the true underlying distribution of the toy problem. On the left, we have the sampled training data.

Figure 2: In this figure, we show that even though the model becomes much more expressive as we increase the number of parameters, as shown in the possible decision boundaries of the poisoned network of various sizes, *both SGD and Guess & Check produce decision boundaries that are relatively stable* as we increase the number of parameters. From left to right, we have decision boundaries produced by 2, 4, 10, 15, 20 hidden units single layer neural networks with different training methods. For each (training method, model size) pair, we show 9 randomly sampled decision boundaries of the trained network. We showed even more samples of the decision boundaries in Appendix A

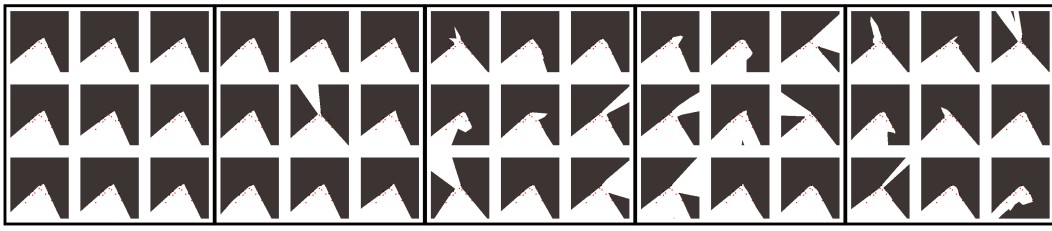

(a) Decision boundaries of a poisoned neural network



(b) Decision boundaries of SGD trained models

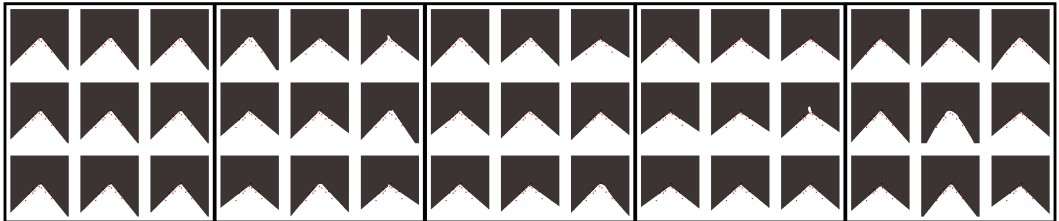

(c) Decision boundaries of Guess & Check trained models

## 3.1 OVERPARAMETERIZATION INCREASES MODEL COMPLEXITY

To illustrate how the model complexity increases with number of parameters, we first poison a model by minimizing the loss on the training data while maximizing the loss on the testing data. Given we only examine cases where the model (trained by SGD) achieves 100% training accuracy, this represents the worst-case decision boundary for the unpoisoned loss. As we increase the number of hidden units from 2 to 20, the decision boundary becomes much more ill-behaved (see Figure 2a). When the hypothesis class is restricted to 2 hidden units (the left most plots in Figure 2a), the model can only fit the data by using a single kink in the decision boundary, so it has to trade off either fitting the training examples to 100% accuracy or performing badly on the poison objective. Given that the model fits training data, it has to perform well on testing data. This is consistent with the under-parameterized regime and with classical learning theory.

As we increase the model size, the model class now contains strange decision boundaries that can fit the training data while performing poorly on the testing data. From the perspective of classical learning theory, we would expect models with 20 hidden units to perform much worse than the model with 2 hidden units. Surprisingly, even though more complicated decision boundaries are available as we increase the model size, we never see such boundaries when optimizing the (unpoisoned) training objective with SGD. For example, in Figure 2b, as we increase the number of hidden units, the decision boundary remains relatively consistent. Given that both the weird and nice decision boundaries exist in the model class, it is natural to ask what biases the learned network towards good vs. bad optima.

Due to the consistent behavior of SGD trained networks in the overparameterized regime, it is only reasonable that people started investigating gradient dynamics as a source of implicit regularization (Galanti & Poggio, 2022; Arora et al., 2019; Advani et al., 2020; Liu et al., 2020). However, we show

below that, rather surprisingly, we can obtain similar generalization behavior on the toy problem by using a Guess & Check algorithm that is completely free of gradient dynamics.

## 3.2 GENERALIZING ON TOY PROBLEM WITHOUT GRADIENTS

In our toy setting, we find that generalization is surprisingly generic with respect to the dynamics of optimizers. To avoid using optimizers with the same inductive biases as gradient methods, we experiment with *Guess & Check*: we repeatedly sample parameters until a model achieves 100% training accuracy with train loss below a certain threshold. Unlike other optimizers, we do not use any gradient information, and we also do not take any iterative steps. Surprisingly, even with Guess & Check, we often end up with a well-behaving decision boundary like the one that we trained with regular SGD, see Figure 2c.

From the simple two class toy problem, we can see clearly that Guess & Check solutions already endow the learned model with a very strong implicit bias that does not originate from gradient dynamics. In the next section, we extend a similar analysis to common datasets such as MNIST & CIFAR10 to see whether this observation continues to hold in more practical settings.

## 4 EXPERIMENTS

### 4.1 NON-GRADIENT BASED OPTIMIZERS

In our experiments, we test three different non-gradient based optimizers: Guess & Check, Pattern Search, and Greedy Random Search on varying scales of MNIST & CIFAR-10 and on different architectures. Here, we explain each optimizer.

#### 4.1.1 GUESS & CHECK

The Guess & Check algorithm optimizer randomly generates parameter vectors with entries sampled independently and uniformly from $[-1, 1]$[1]. If the randomly sampled model achieves 100% training accuracy and has training loss below a chosen threshold, then the model is kept and the optimizer terminates. If not, the vector is thrown away and we keep guessing new vectors until our conditions are met.

Guess & Check is of theoretical value because its only implicit bias comes from the geometry of the loss landscape, and its success implies the volume hypothesis. With this optimizer, the likelihood that a set of solutions are selected is exactly proportional to the volume of the set in parameter space. If a model consistently generalizes well when trained with Guess & Check, then this means the set of "good" minima has large volume among low-loss parameters. We do want to make a distinction between flat solutions (Keskar et al., 2016) and solutions with large volumes. It is possible that a collection of solutions has a very large volume but is not itself a flat basin but rather a collection of many small volume regions that have large volume in aggregate.

When we train with Guess & Check, we can be confident that gradient-based implicit regularization plays no role in the final performance – the volume hypothesis is the only source of implicit regularization. Unfortunately, Naïve guess-and-check suffers from the problem that the cost of interpolating the training data grows exponentially as the number of training examples or classes increase, so we have restricted experiments with Guess & Check to few-shot problems with smaller sample sizes.

#### 4.1.2 LOCAL NON-GRADIENT BASED OPTIMIZER

Due to the difficulty of scaling Guess & Check to large problems, we explore two alternative non-gradient based optimizers, Pattern Search and Random Greedy Search, that work for bigger datasets. Like SGD, both approaches update the model using a local search and may have biases that originate from factors other than volume alone. The success of Pattern Search and Random Greedy demonstrates that gradient optimization is not *strictly* needed to observe implicit regularization, but they may exploit regularization properties of local search that are also exploited by SGD.

**Pattern Search**   Pattern Search randomly selects a parameter in the model and takes a step of fixed size that is randomly chosen to be either positive or negative, if the model achieves lower loss after

---

[1]See Appendix C for experiments with other sampling intervals.

Table 2: On the two class MNIST problem, G&C performs comparably to SGD across different train loss level and number of samples. This shows us that despite the large number of parameters, G&C solutions are implicitly regularized. To show that the degree of generalization of G&C is indeed substantial, we train an additional linear model, which has a much more restricted hypothesis class, but has on average 10% worse generalization performance. The empty cells correspond to linear models where we could not find solutions with 100% training accuracy. We also show the estimated standard deviations of the averages computed over 175 random data split and training seeds. For most cells, the standard deviation is less than 1%.

| Sample Count | Arch | Optimizer | Best Test Acc | Train Loss (0.3, 0.35) | (0.35, 0.4) | (0.4, 0.45) | (0.45, 0.5) | (0.5, 0.55) | (0.55, 0.6) | (0.6, 0.65) |
|---|---|---|---|---|---|---|---|---|---|---|
| 32 | LeNet | G&C | 93.02%±0.27% | 93.02%±0.27% | 92.39%±0.29% | 90.59%±0.34% | 89.18%±0.38% | 87.22%±0.43% | 86.23%±0.44% | 83.15%±0.51% |
|  | LeNet | SGD | **94.04%±0.25%** | - | - | 94.04%±0.25% | 93.49%±0.28% | 92.54%±0.28% | 91.63%±0.33% | 88.60%±0.35% |
|  | Linear | SGD | 84.75%±0.47% | 84.75%±0.47% | 82.69%±0.43% | 81.24%±0.44% | 79.04%±3.14% | 78.94%±4.74% | - | - |
| 16 | LeNet | G&C | 89.21%±0.47% | 89.21%±0.47% | 87.01%±0.50% | 85.18%±0.56% | 84.69%±0.54% | 81.91%±0.62% | 78.61%±0.65% | 75.37%±0.63% |
|  | LeNet | SGD | **91.24%±0.40%** | 91.24%±0.40% | 90.87%±0.41% | 90.84%±0.38% | 88.77%±0.48% | 87.93%±0.48% | 86.98%±0.47% | 83.90%±0.49% |
|  | Linear | SGD | 80.68%±0.55% | 80.68%±0.55% | 78.50%±0.56% | 75.69%±0.60% | 72.09%±0.56% | 67.16%±0.67% | 69.51%±3.40% | - |
| 8 | LeNet | G&C | 83.05%±0.67% | 83.05%±0.67% | 80.72%±0.75% | 78.23%±0.81% | 78.05%±0.72% | 76.40%±0.79% | 70.76%±0.74% | 67.48%±0.78% |
|  | LeNet | SGD | **84.82%±0.63%** | 83.63%±0.63% | 84.82%±0.63% | 82.62%±0.74% | 81.85%±0.72% | 79.70%±0.70% | 79.74%±0.63% | 76.51%±0.71% |
|  | Linear | SGD | 74.29%±0.72% | 74.29%±0.72% | 71.72%±0.75% | 67.79%±0.69% | 67.36%±0.76% | 63.46%±0.75% | 58.65%±0.79% | 54.87%±0.75% |
| 4 | LeNet | G&C | 76.28%±0.90% | 76.28%±0.90% | 73.93%±0.92% | 72.63%±0.86% | 70.89%±0.90% | 68.27%±0.83% | 65.63%±0.92% | 62.38%±0.91% |
|  | LeNet | SGD | **77.35%±0.81%** | 77.35%±0.81% | 75.01%±0.85% | 75.61%±0.83% | 73.95%±0.85% | 73.28%±0.85% | 69.15%±0.84% | 67.65%±0.84% |
|  | Linear | SGD | 65.12%±0.81% | 65.12%±0.81% | 61.94%±0.82% | 62.14%±0.78% | 58.11%±0.88% | 57.21%±0.91% | 55.38%±0.88% | 53.60%±0.83% |
| 2 | LeNet | G&C | 66.89%±1.04% | 66.89%±1.04% | 65.87%±1.05% | 64.03%±0.92% | 62.81%±0.90% | 61.02%±0.84% | 59.90%±0.91% | 56.82%±0.95% |
|  | LeNet | SGD | **69.67%±0.98%** | 69.67%±0.98% | 67.11%±0.93% | 64.94%±0.95% | 63.42%±0.87% | 64.38%±0.88% | 63.82%±0.89% | 62.33%±0.87% |
|  | Linear | SGD | 58.93%±0.94% | 58.93%±0.94% | 58.45%±0.92% | 56.59%±0.89% | 54.11%±0.91% | 54.21%±0.87% | 53.13%±0.93% | 51.59%±0.89% |

taking the step, then the parameter is accepted as the new starting point. If Pattern Search fails to find a step that decreases the loss after going through all the parameters, then the step size is decreased by a constant factor. We repeat this procedure until a solution is found that achieves 100% training accuracy. In our experiments, we use a starting radius of 1, and we decrease the radius by a factor of 2 when it fails to find a descent direction.

**Random Greedy Search** Random Greedy Search adds Gaussian noise to the initial parameter vector with standard deviation of $\sigma$. If the noised solution improves training loss, then the noised solution is accepted as a new starting point. If no solution is found after a fixed number of steps, then $\sigma$ is decreased by a chosen factor before the search continues. Again, we repeat this procedure until a parameter is found that achieves 100% training accuracy. In our experiment, we start the procedure with $\sigma = 1$. If we fail to find a perturbation that decreases loss after 30000 random steps, then we decrease $\sigma$ by a factor of 2.

### 4.2 RESULTS ON 2-CLASS CIFAR-10/MNIST

In this section, we apply the Guess & Check algorithm on a conventional LeNet model on MNIST (LeCun et al., 2010) and CIFAR-10 (Krizhevsky, 2009). Due to the exponential time complexity of the Guess & Check algorithm, we stick with 2-class problems with fewer than 32 total training samples. To enable fair comparisons between G&C and SGD optimized models, we compare the performance of the models across different train loss levels after the model's weights have been normalized. This is crucial for a fair comparison because it has been observed that lower loss levels corresponds to better generalization even after train accuracy has reached 100%.

We find that given the same loss level and number of samples, Guess & Check performs comparably to SGD, especially at lower loss levels. In the case of CIFAR-10, Guess & Check even outperforms SGD solutions by a substantial margin. This result is made even more interesting given that the models are capable of pathological overfitting: they are able to completely misclassify the validation set while achieving 100% training accuracy (see Table 1).

| MNIST | | CIFAR | |
|---|---|---|---|
| # Samples | Val. Acc. | # Samples | Val. Acc. |
| 32 | 0% | 24 | 0% |
| 16 | 0% | 16 | 0% |
| 8 | 0% | 8 | 0% |
| 4 | 0% | 4 | 0% |
| 2 | 0% | 2 | 0% |

Table 1: Comparing poisoned validation error. In this table, we attempt to fit the training data of various sizes while poisoning LeNet with the wrong validation labels. We find that the LeNet we use is of sufficient capacity that it can completely fit the training data while failing to classify the validation set.

To illustrate how well G&C performs, we also train a linear model on MNIST for comparison. Even though the hypothesis class is now restricted to only linear solutions, a significantly smaller hypothesis class compared to LeNet, the linear model still underperforms the Guess & Check solution on LeNet by more than 10% in many cases. Note that, despite being convex, solutions of the linear problem vary because we use SGD and apply early stopping when the desired loss level is achieved.

Table 3: On the two class CIFAR10 problem, G&C performs comparably to SGD across different training losses and numbers of samples. This shows us that, despite the large number of parameters, G&C solutions are implicitly regularized. We do note that G&C in this low data regime consistently performs better than SGD though. We computed the standard deviation over 75 random data splits and training seeds.

| Sample Count | Optimizer | Best Test Acc | Train Loss (0.55, 0.57) | (0.57, 0.59) | (0.59, 0.61) | (0.61, 0.63) | (0.63, 0.65) | (0.65, 0.67) |
|---|---|---|---|---|---|---|---|---|
| 24 | G&C | **66.59%±0.74%** | 66.59%±0.74% | 65.91%±0.80% | 64.09%±0.96% | 61.08%±0.89% | 59.33%±0.88% | 57.18%±0.89% |
|  | SGD | 63.16%±0.87% | 63.16%±0.87% | 62.02%±0.84% | 60.74%±0.73% | 58.21%±0.75% | 57.62%±0.69% | 56.24%±0.55% |
| 16 | G&C | **61.10%±0.98%** | 61.10%±0.98% | 59.54%±0.98% | 59.21%±0.90% | 57.53%±0.86% | 57.71%±0.81% | 55.06%±0.70% |
|  | SGD | 58.98%±0.69% | 58.58%±0.77% | 58.98%±0.69% | 57.86%±0.79% | 57.11%±0.61% | 56.77%±0.62% | 53.90%±0.50% |
| 8 | G&C | **57.17%±0.94%** | 54.39%±0.80% | 53.99%±0.76% | 57.17%±0.94% | 54.61%±0.68% | 52.66%±0.66% | 52.82%±0.62% |
|  | SGD | 56.76%±0.71% | 56.76%±0.71% | 55.02%±0.62% | 54.79%±0.72% | 54.62%±0.68% | 53.39%±0.66% | 53.53%±0.55% |
| 4 | G&C | **55.51%±0.84%** | 55.51%±0.84% | 53.59%±0.96% | 52.78%±0.82% | 52.30%±0.67% | 52.38%±0.63% | 54.07%±0.72% |
|  | SGD | 53.75%±0.62% | 53.49%±0.68% | 52.14%±0.51% | 53.75%±0.62% | 51.53%±0.63% | 52.18%±0.66% | 50.44%±0.55% |
| 2 | G&C | **52.39%±0.67%** | 51.66%±0.74% | 52.39%±0.67% | 52.00%±0.60% | 51.37%±0.56% | 50.01%±0.71% | 50.66%±0.62% |
|  | SGD | 51.98%±0.59% | 51.93%±0.66% | 51.39%±0.47% | 51.98%±0.59% | 51.16%±0.48% | 50.65%±0.45% | 50.05%±0.43% |

Table 4: In this table, we trained the same LeNet, but with more examples and more classes. We found that the generalization performance is still fairly similar between SGD and alternative zeroth order optimizers that do not use any gradient information. The empty cells indicate the experiment has timed out, and we failed to find models achieving 100% training accuracy within a reasonable time limit.

|  | Sample Count | 1000 | 500 | 300 | 100 |
|---|---|---|---|---|---|
| MNIST | SGD | 93.46%±0.11% | 90.15%±0.22% | **87.48%±0.26%** | **78.67%±0.51%** |
|  | Pattern Search | **93.68%±0.12%** | 90.33%±0.12% | 87.26%±0.30% | 78.43%±0.46% |
|  | Random Greedy | 93.34%±0.08% | **90.35%±0.10%** | 87.33%±0.21% | 78.51%±0.50% |
| CIFAR-10 | SGD | **36.01%±0.25%** | 29.91%±0.31% | **25.88%±0.34%** | **19.86%±0.27%** |
|  | Pattern Search | - | **30.00%±0.69%** | 25.04%±0.66% | 18.70%±1.22% |
|  | Random Greedy | 34.44%±0.54% | 27.06%±0.75% | 24.04%±0.58% | 16.80%±0.13% |

Even though the number of samples is small, we do note that this regime highlights the effects of overparametrization. For example, in our LeNet for MNIST, we have 11074 parameters, which is orders of magnitude larger than the number of examples. Yet the model continues to generalize well relative to SGD, showing us that the large volume of the good solution set is on its own enough to bias the optimizer towards favorable generalization.

Even though the generalization performance is similar between the SGD and G&C solutions, we do note that the test accuracies are not exactly the same between models trained with both methods, implying that SGD may have additional bias that G&C does not take into account. However, our main argument is that optimizer-specific bias is not needed to explain generalization, and may not even be the primary cause of generalization behavior; in our experiments here, the bulk of generalization can be explained by the geometry of the loss landscape.

## 4.3 RESULTS ON 10-CLASS CIFAR-10/MNIST

In this section, we evaluate the importance of gradient-based optimizers in the setting where more classes are involved. However, the Guess & Check algorithm is no longer feasible due to the exponential time complexity. Instead of Guess & Check, we employ Pattern Search and Greedy Random Search to evaluate the dependence of generalization on gradient based optimizers. Again, we find that these non-gradient based optimizers offer similar levels of generalization benefits as SGD despite not using any gradient information at all.

In Table 4, we see that Greedy Random Search and Pattern Search both generalize comparably to SGD. The average performance difference is only 0.9% across different sample sizes, datasets, and optimizer combinations. In several cases, Pattern Search even performs better than SGD. Even though 0.9% may seem large when viewed from the perspective of achieving state-of-the-art accuracy, we note that a performance difference of 0.9% is within the margin that can be expected from hyperparameter tuning, and that we have not tuned either of the zeroth order optimizers, and yet they still achieve a comparable level of generalization to SGD.

| | Loss Level | | | | | | |
|---|---|---|---|---|---|---|---|
| width | (0.3, 0.35) | (0.35, 0.4) | (0.4, 0.45) | (0.45, 0.5) | (0.5, 0.55) | (0.55, 0.6) | (0.6, 0.65) |
| 1 | n/a | n/a | n/a | n/a | n/a | 92.11%±1.35% | 93.44%±n/a |
| 0.9 | n/a | n/a | n/a | n/a | 90.89%±n/a | 86.33%±6.81% | 90.90%±3.03% |
| 0.8 | n/a | n/a | 93.34%±n/a | 83.89%±8.97% | 87.71%±n/a | 93.29%±n/a | 81.94%±7.35% |
| 0.7 | 97.73%±n/a | 96.32%±n/a | 95.20%±0.42% | 92.30%±1.21% | 86.10%±6.24% | 80.59%±2.96% | 87.37%±n/a |
| 0.6 | 91.20%±1.40% | 89.66%±1.88% | 87.76%±2.22% | 85.45%±2.07% | 83.64%±2.78% | 82.76%±3.33% | 79.43%±3.06% |
| 0.5 | 89.21%±0.47% | 87.01%±0.50% | 85.18%±0.56% | 84.69%±0.54% | 81.91%±0.62% | 78.61%±0.65% | 75.37%±0.63% |
| 0.4 | 85.82%±0.53% | 83.78%±0.61% | 81.43%±0.61% | 79.63%±0.65% | 77.50%±0.64% | 75.98%±0.66% | 72.30%±0.77% |
| 0.3 | 79.55%±0.60% | 79.13%±0.60% | 76.63%±0.72% | 75.43%±0.63% | 74.22%±0.66% | 72.28%±0.72% | 71.75%±0.67% |
| 0.2 | 77.39%±0.56% | 76.08%±0.63% | 74.70%±0.56% | 73.40%±0.60% | 71.84%±0.60% | 69.66%±0.64% | 68.49%±0.58% |

Table 6: Performance of G&C on MNIST with 16 samples as we scale up the model. The n/a indicates that a model has not been found for the cell.

## 4.4 Few-Shot Learning with ResNets

In this section, we evaluate the importance of gradient-based optimizers in the few-shot setting. This setting enables us to compare gradient methods to zeroeth order optimization using industrial-scale models. For the most part, we find that the gradient-free Pattern Search optimizer performs comparably to SGD in the 1-shot setting.

Few-shot learning is usually used to test the ability of models to generalize to unseen tasks given limited training examples. This is a perfect evaluation task for our hypothesis for the following reasons: First, in few-shot learning, we only use 1 or 5 training images per class during the evaluation stage, which makes zeroth order optimization possible. Second, few-shot learners usually utilize a pre-trained feature extractor on the base classes, and only learn a new classification head by SGD or other solvers such as SVM and ridge regression (Lee et al., 2019; Bertinetto et al., 2019) given the unseen tasks. This setting limits the dimension of learnable parameters, thus making training deeper networks such as ResNet possible with these non-gradient based optimizers. Finally, although we only attempt to learn a single layer, due to the few training examples (1 or 5 per class), we will still have an overparameterized model, which is the setting we are interested in.

We evaluate the effectiveness of non-gradient based optimizers on CIFAR-FS (Bertinetto et al., 2019) and mini-ImageNet (Vinyals et al., 2016) with ResNet-12, a commonly used architecture in the few-shot classification literature. During the training stage, we pre-train a feature extractor on the base classes and evaluate the generalization on unseen tasks via 1-shot-5-way episodes, where each episode is a 5-way classification problem and each class contains 1 training image. During the evaluation stage, given the unseen episodes and pre-trained feature extractor, we learn a new classification head with a specific optimizer and evaluate the performance on the testing images. We compare the testing accuracy on 600 different episodes between SGD, and Pattern Search in Table 5. For Pattern Search, instead of stopping optimization immediately after fitting all the training examples, we keep updating the model until $t$ steps, where we set $t = 3000$ for both CIFAR-FS and mini-ImageNet. As showed in Table 5, Pattern Search always outperforms SGD by a large margin, i.e., over 2% for both CIFAR-FS and mini-ImageNet, which suggests that gradient-based optimizers are not necessary in the few-shot setting.

Table 5: 1-shot-5-way classification performance on both CIFAR-FS and mini-ImageNet with ResNet-12 backbone. We provide mean test accuracy over 600 episodes and the one standard error. Compared to SGD, Pattern Search can always achieve better performance by a large margin.

| Optimizer | CIFAR-FS | mini-ImageNet |
|---|---|---|
| SGD | $68.35 \pm 0.46$ | $55.76 \pm 0.42$ |
| Pattern Search | $\mathbf{70.25 \pm 0.45}$ | $\mathbf{58.53 \pm 0.41}$ |

## 5 How does G&C behave as we scale up the model?

People have observed that increasing the size of neural networks trained with SGD can lead to either double descent behavior Nakkiran et al. (2021) or increasing performance Kaplan et al. (2020). This phenomenon has been previously attributed to the regularization effect of SGD. However, given the similarity of performance between G&C models and SGD-trained models, it is natural to ask whether we observe similar behaviors in G&C models as we increase the number of parameters. Here, we show that G&C models continue to improve as we increase their size, without using SGD.

To investigate this further, we conducted experiments on 2-class MNIST using G&C with varying widths of LeNet (Table 6). Surprisingly, we found that as we increased the width of the model, its validation accuracy also increased. This observation contradicts generalization theories based on model capacity, which suggest that increasing model size beyond a certain point leads to overfitting and reduced generalization performance.

The observation points to the hypothesis that increasing the width of a neural network can expand the volume of the good function class. If we can identify the specific function within this class that experiences an increase in volume with more parameters, it may be possible to achieve similar benefits with a smaller model that captures the favorable properties of the function. This could lead to more efficient and effective deep learning models that perform better without unnecessary parameter bloat. However, further research is required to investigate this hypothesis and identify the specific functions that contribute to the observed increase in volume.

# 6 A Toy Example: Simplicity bias may originate from the volume bias as opposed to SGD

In this section, we study whether volume may explain the simplicity bias previously observed in Shah et al. (2020).

While we have mostly measured bias in terms of generalization in this paper, we think it is a promising future direction to quantify whether other forms of bias can be attributed to the volume hypothesis instead. One example is simplicity bias, where trained neural networks strongly prefer linear decision boundaries compared to robust ones. We provide a toy illustration on why this may be attributed to the volume hypothesis, but leave further exploration of this as future work.

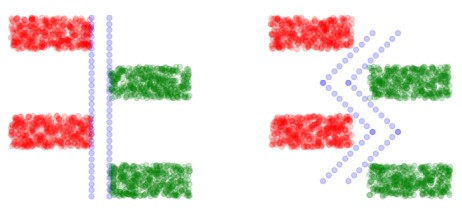

Figure 3: The volume of the decision boundary on the left as measured by G&C is $10^{-4}$ whereas the volume of the robust/complex decision boundary has volume smaller than $10^{-10}$. The large volume disparity may explain trained network's strong preference for the linear solution.

Consider the following example: a trained neural network on the slab dataset ignores the more complex y-axis, as shown on the left of Figure 3, and uses a linear decision boundary drawn along the x-axis only as opposed to the robust decision boundary shown on the right of Figure 3. While Shah et al. (2020) has attributed the simplicity bias in this example to SGD, we found that the simplicity bias may simply originate from the large disparity in volumes between the linear and robust functions in the loss landscape. In fact, when we used G&C to measure the volume of the two respective decision boundaries in the parameter space, we found that the linear decision boundary has volume that is 6 orders of magnitude larger than that of the robust decision boundary. Specifically, we estimate the volume of the solution by taking the reciprocal of the number of guesses before a solution is obtained. The volume disparity may explain why the simple decision boundary is strongly preferred compared to the alternative.

# 7 Conclusion

In this paper, we empirically show that gradient-based implicit regularization of training dynamics is not required for generalization. Instead, we consider non-gradient optimizers that lack gradient dynamics, yet still perform well. The strong performance of gradient-free optimizers, in particular Guess & Check, strongly suggests that the disparate volume of good and bad hypothesis classes is the main implicit bias that enables these optimizers to succeed. For future work, we think more critically examining the role of volume as implicit bias in neural networks will be a fruitful and interesting direction.

## 8 REPRODUCIBILITY STATEMENT

We ran all the experiments with prespecified random seeds, so all of the tables will be reproducible by running the respective scripts in our included code base.

## 9 ACKNOWLEDGEMENTS

This work was supported by the ONR MURI program, the National Science Foundation (IIS-2212182), the AFOSR MURI Program, DARPA GARD (HR00112020007), and the Office of Naval Research. Further support was provided by Capital One Bank.

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

## A  ADDITIONAL EXPERIMENTS FOR TOY EXAMPLES

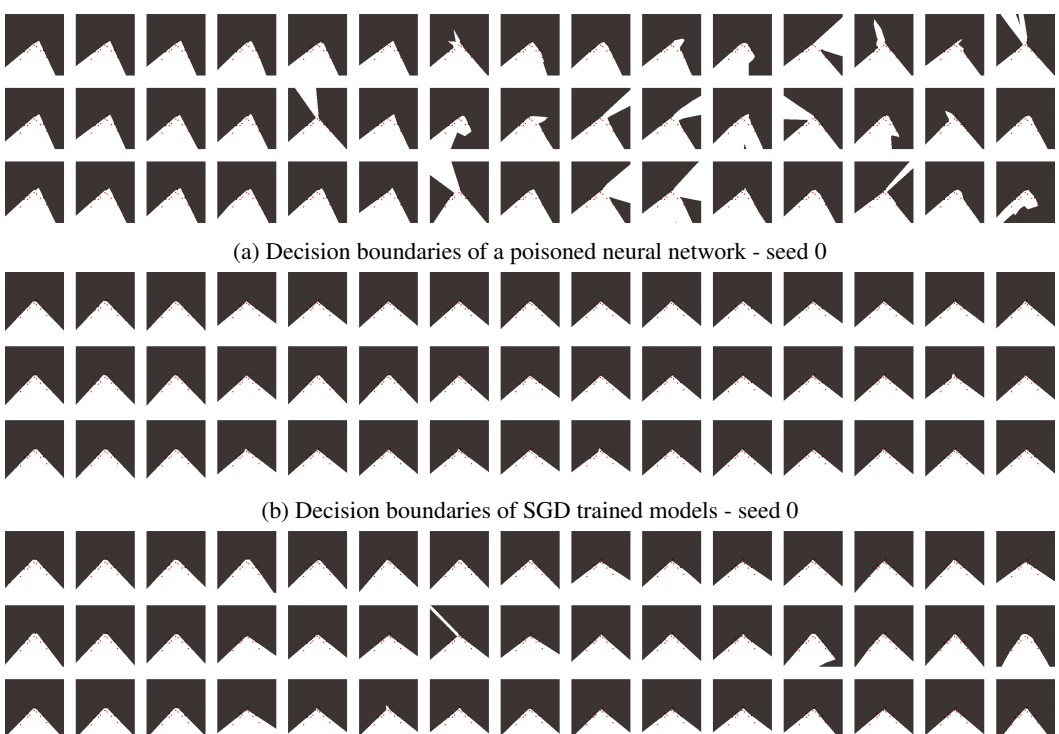

(a) Decision boundaries of a poisoned neural network - seed 0

(b) Decision boundaries of SGD trained models - seed 0

(c) Decision boundaries of Guess & Check trained models - seed 0

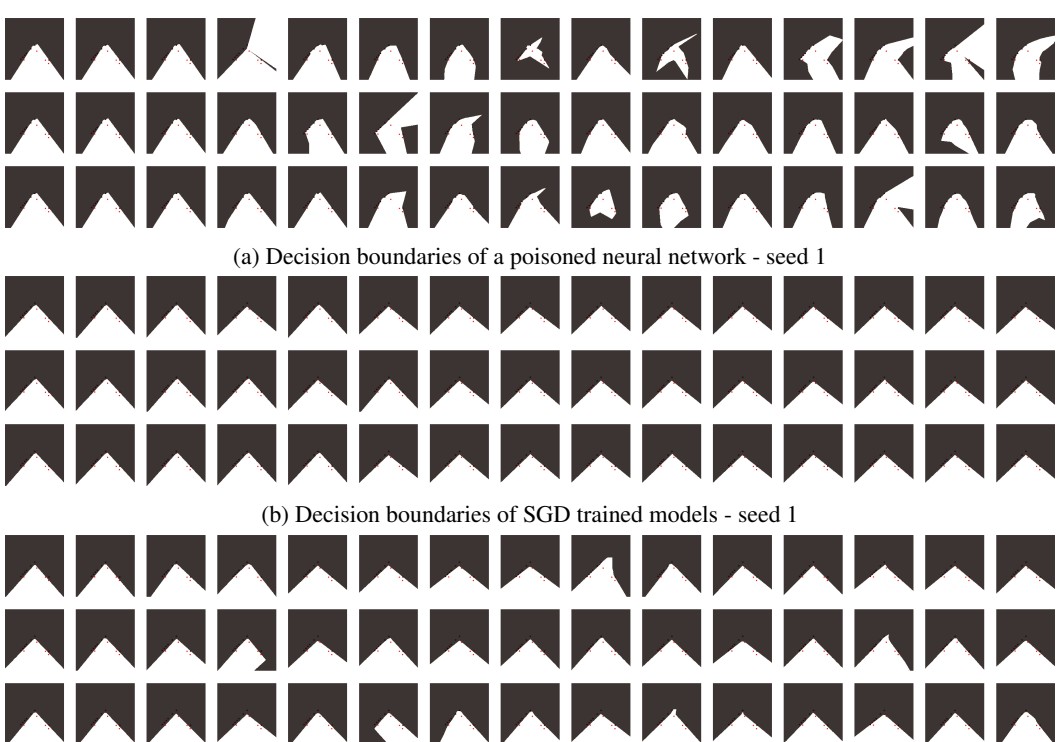

(a) Decision boundaries of a poisoned neural network - seed 1

(b) Decision boundaries of SGD trained models - seed 1

(c) Decision boundaries of Guess & Check trained models - seed 1

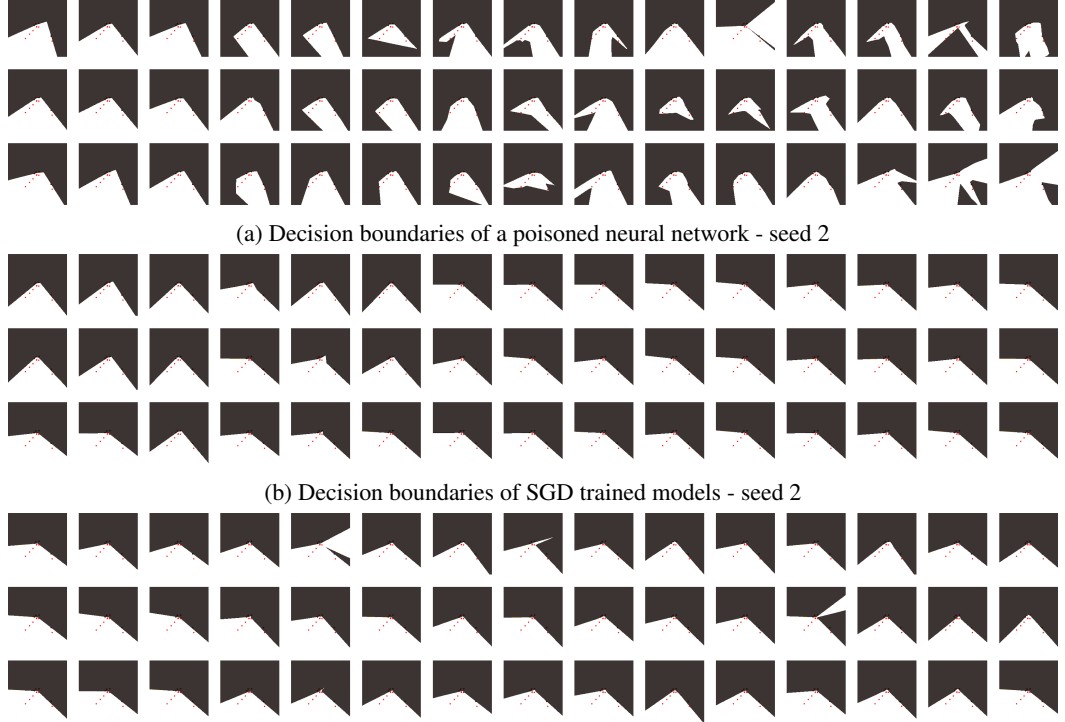

(a) Decision boundaries of a poisoned neural network - seed 2

(b) Decision boundaries of SGD trained models - seed 2

(c) Decision boundaries of Guess & Check trained models - seed 2

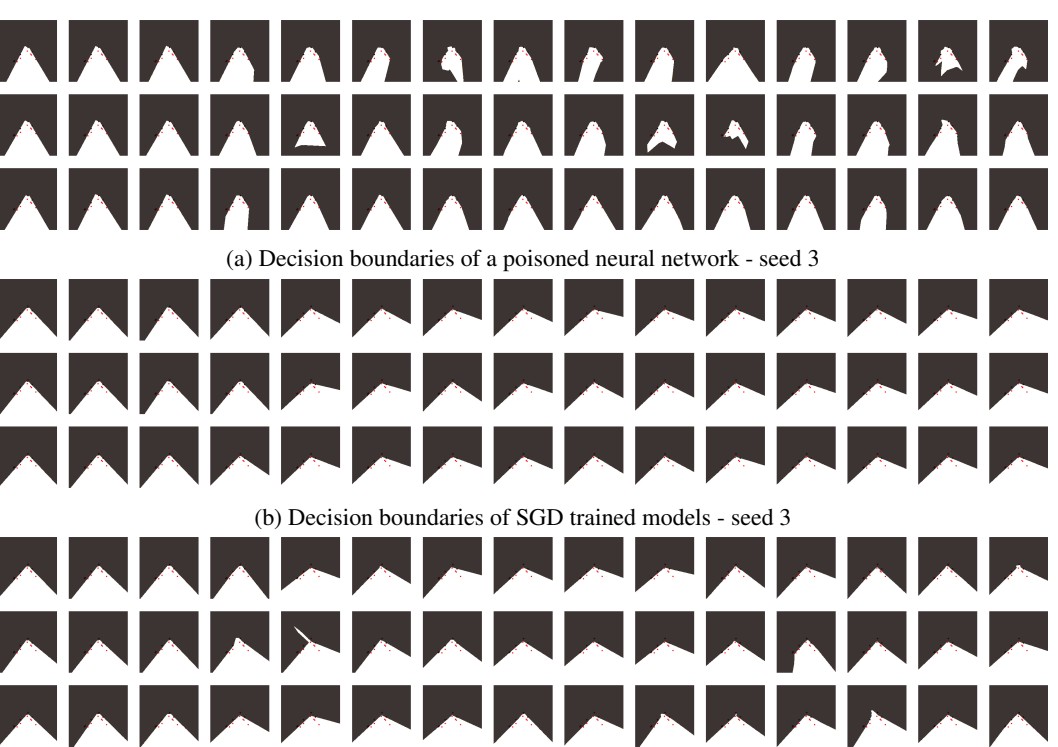

(a) Decision boundaries of a poisoned neural network - seed 3

(b) Decision boundaries of SGD trained models - seed 3

(c) Decision boundaries of Guess & Check trained models - seed 3

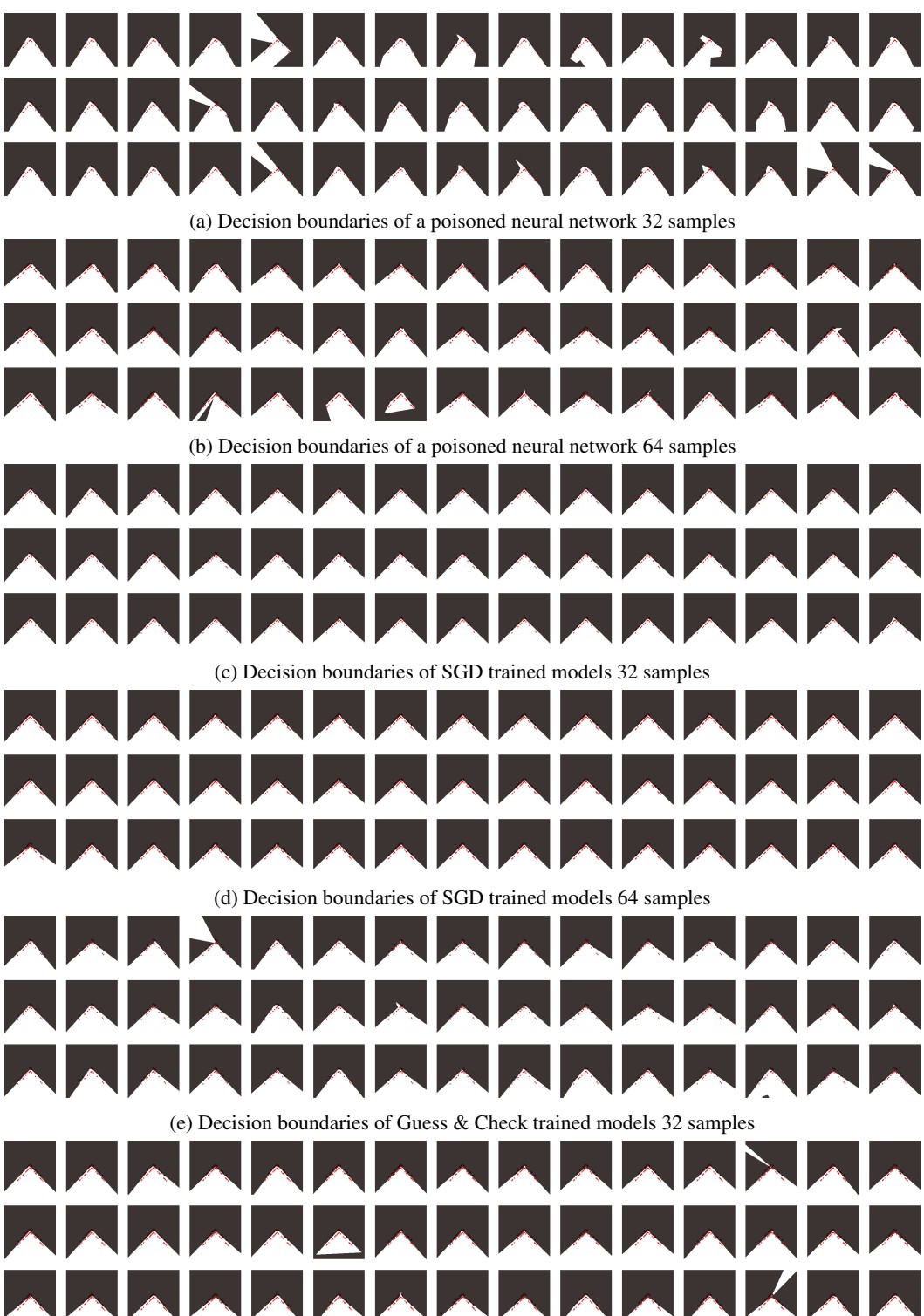

(a) Decision boundaries of a poisoned neural network 32 samples

(b) Decision boundaries of a poisoned neural network 64 samples

(c) Decision boundaries of SGD trained models 32 samples

(d) Decision boundaries of SGD trained models 64 samples

(e) Decision boundaries of Guess & Check trained models 32 samples

(f) Decision boundaries of Guess & Check trained models 64 samples

Figure 8: Decision boundaries as we increase the number of samples

# B  INFLUENCE OF SAMPLING RANGE FOR GUESS & CHECK OPTIMIZER

|  | | Loss Level | | | | | | |
| --- | --- | --- | --- | --- | --- | --- | --- | --- |
|  | L2 Norm | (0.3, 0.35) | (0.35, 0.4) | (0.4, 0.45) | (0.45, 0.5) | (0.5, 0.55) | (0.55, 0.6) | (0.6, 0.65) |
| Uniform (-1,1) | 60 | **93.02%** | **92.39%** | 90.59% | 89.18% | 87.22% | 86.23% | **83.15%** |
| Uniform (-2,2) | 120 | 92.52% | 90.16% | 89.06% | **89.89%** | 87.89% | 85.17% | 82.17% |
| Uniform (-5,5) | 300 | 92.79% | 92.04% | **91.18%** | 87.59% | 87.16% | 86.42% | 82.40% |
| Sphere | 100 | 92.87% | 92.20% | 89.79% | 89.84% | **87.91%** | **86.70%** | 82.81% |

Table 7: Comparing performance of models given different sampling methods on MNIST with 24 samples. We tested uniform sampling between (-2, 2) and (-5, 5). We also tested sampling from a sphere with $\mathcal{L}_2$ norm of 100. For the most part, we found that the sampling range does not materially change the performance of the model.

|  | | Loss Level | | | | | |
| --- | --- | --- | --- | --- | --- | --- |
|  | L2 Norm | (0.55, 0.57) | (0.57, 0.59) | (0.59, 0.61) | (0.61, 0.63) | (0.63, 0.65) | (0.65, 0.67) |
| Uniform (-1,1) | 60 | 66.59% | **65.91%** | 64.09% | 61.08% | 59.33% | 57.18% |
| Uniform (-2,2) | 120 | 65.93% | 65.14% | **64.93%** | 61.31% | **61.85%** | 57.22% |
| Uniform (-5,5) | 300 | **67.63%** | 61.69% | 63.63% | **62.19%** | 60.33% | 58.41% |
| Sphere | 100 | 61.81% | 64.31% | 59.30% | 62.01% | 56.72% | **60.77%** |

Table 8: Comparing performance of models given different sampling methods on CIFAR with 36 samples. We tested uniform sampling between (-2, 2) and (-5, 5). We also tested sampling from a sphere with $\mathcal{L}_2$ norm of 100. For the most part, we found that the sampling range does not materially change the performance of the model.

