# OpenReview forum: "Loss Landscapes are All You Need: Neural Network Generalization Can Be Explained Without the Implicit Bias of Gradient Descent"
_ICLR.cc/2023/Conference — ICLR 2023 notable top 25%_

### Official Review · Reviewer_rv2f · 2022-10-20

**Confidence:** 4
**Correctness:** 3
**Technical Novelty And Significance:** 4
**Empirical Novelty And Significance:** 4
**Recommendation:** 8

**Clarity, Quality, Novelty And Reproducibility:**

## Clarity

Overall, the paper is easy to follow.

However, the readability could be improved:
* Figure 2: the colors chosen by the authors are absolutely identical when printed in black and white, colors with different luminosity would be appreciable;
* Table 1: "Linear" is not the name of an optimizer, it is the model. It would be clearer to write "Linear model" instead of "Linear". Also, the label of the column could be changed into "Optimizer/Model", for instance;
* the equation where the function $f(x) = a x^2 + \sum_i b_i x + c$ is defined should be numbered.

## Novelty

Research about the "implicit bias" of the SGD when training neural networks (NNs) is still a hot topic: obtaining theoretical results in this field would be a step forward in the understanding of generalization in NNs.

In this paper, the authors challenge this approach by showing that it remains possible to obtain good generalization results when training a NN with zero-order methods, such as "Guess & Check". This kind of results should help other researchers to find a good approach of generalization results, which seem not to be specific to gradient-related methods.

## Quality

In spite of the narrowness of the experiments (small NNs, very small training datasets), generalization results about the "Guess & Check" method shown in Tables 1 and 2 seem to be significant, since the authors report the evolution of the train loss *after the train accuracy has attained 100%*, trey tried different training set sizes, and they compared their results to a simple linear model. More surprisingly, "Guess & Check" outperforms SGD on a subset of CIFAR-10.

However, the relevance of the other two methods ("Pattern Search" and "Random Greedy Search") is questionable and should have been discussed. These methods are very close to a SGD with injection of noise. Besides, they are technically very close to each other. In short, measuring a difference of loss between two parameters that are very close to each other is almost the same as computing the gradient with some noise. So, I am not sure that these methods can fairly be called "gradient-free methods" in this context (computationally, it is true, since they do not require a full computation of the gradient, but they contain an informal, empirical computation of the gradient anyway).

In Section 5, several points are too vague:
1. The notion of "volume" is not defined. Actually, if we use the standard definition, then it is obvious that the volume of a subspace of the space of parameters is zero. So, there is no point in comparing the volumes of such subspaces.
2. The proposed function $f(x) = a x^2 + \sum_i b_i x + c$ is not similar to the NNs, from the authors' point of view. When we add parameters in a NN, the relative "volume" of some subsets of the model space may vary, but the whole space of functions that can be represented by the NN varies at the same time (it grows). So, it would be better to simulate this property by changing a bit the proposed function. For instance: $f(x) = \epsilon \sum_{i = 2}^{p} a_i x^i + \sum_{i = 1}^p a_i x + c$, where $\epsilon > 0$ should be very small. That way, the "volume" taken by *almost* linear functions (at least on a compact set) would grow as $p$ grows, and at the same time the expressiveness of the model would grow too, as in a NN.

## Reproducibility

The authors seem to have provided all necessary information to reproduce the results.

## Additional comments

Table 1. As such, it is difficult to comment the validation loss, since there is no point of comparison (e.g., a model achieving the best or the worst validation loss possible, with the same NN architecture). It would have been interesting to add the results of a "poisoned" NN: provided that the train accuracy is 100%, what is the worst achievable validation accuracy? If the results are close to the current results, then we would know that, by design, the model cannot overfit. So, the provided results would be quite unsurprising, and not compatible with existing results for large NNs and datasets (*Understanding deep learning requires rethinking generalization*, Zhang, 2016).

The size of the whole file is around 10 Mb, which is abnormally large, provided that the figures are not so numerous and relatively simple.

**Strength And Weaknesses:**

## Strengths

* This paper challenges a widely spread hypothesis: the generalization power of NNs is due to the "implicit bias" of gradient based methods (e.g., the SGD).
* The comparison between NNs trained by SGD and generated by "Guess & Check", is convincing.
* More specifically, the "Guess & Check" method is non-local, that is, the best vector of parameters is generated by random sampling over the parameter space, and not by refinement of a candidate vector of parameters.

## Weaknesses

* Methods 2 and 3 are very close to the SGD. So, in order to make claims in these cases, it is necessary to have a discussion about this closeness.
* The argument about the volume of a subset of models in the whole model space (see Section 5) is not convincing. Some properties should have been added to the proposed model to make the analogy with NNs more credible.
* Some figures are heavy (slow to load in a PDF reader), and some are unusable when printed in black and white.

**Summary Of The Paper:**

This paper proposes empirical results showing that, when training neural networks, gradient-free optimization methods lead to generalization results comparable to the SGD. So, the "implicit bias" effect, supposed to be specific to the SGD method, should be reconsidered.

Specifically, the authors propose to compare the SGD with 3 other methods:
1. *Guess & Check*: sample randomly the vector of parameters $\theta$, then select the best-performing one (in terms of training loss);
2. *Pattern Search*: at each training step $\theta_t$, select a random direction in the space of parameters, make a step $\delta$ (with given size) in this direction, measure the training loss at this new point $\theta_t + \delta$, then let $\theta_{t+1} = \theta_t + \delta$ if the loss decreases else let $\theta_{t+1} = \theta_t$;
3. *Random Greedy Search*: same as 2, but with random step norm: $||\delta|| \sim |\mathcal{N}(0, \sigma^2)|$.

In methods 2 and 3, the step size decreases when too many steps have been attempted without decreasing the training loss.

**Summary Of The Review:**

This paper sheds new lights to the "implicit bias" phenomenon: the authors have obtained similar generalization results for their gradient-free algorithm and for the SGD. This result should help researchers to understand how generalization works in NNs. But the use of methods 2 and 3 (close to the SGD) is not convincing or even discussed. Section 5 could be largely improved.

EDIT: I acknowledge the improvements done by the authors (mainly in Section 5), so I raise my score.

---

### Official Review · Reviewer_Qs6Q · 2022-10-24

**Confidence:** 4
**Correctness:** 2
**Technical Novelty And Significance:** 2
**Empirical Novelty And Significance:** 2
**Recommendation:** 5

**Clarity, Quality, Novelty And Reproducibility:**

The paper is clear, hypothesis is novel.
Most but not all experiments are fully reproducible.

**Strength And Weaknesses:**

Strengths:
- New and interesting hypothesis that, when properly tested, may attract several investigators towards more useful research directions.

Weaknesses:
Unfortunately the hypothesis has not been tested properly, in my opinion.

- Of the three algorithms, only "Guess and Check" is appropriate for testing the hypothesis.
The other two algorithms ("Patern Search" and "Random Greedy Search"), while they do not require gradient computation, are not fundamentally different from gradient descent since they seek to minimize the loss in a local neighborhood at every step.
Therefore, for the purpose of testing the hypothesis, the results of those two algorithms are irrelevant.

- About the "Guess and Check" algorithm, I consider it strongly limitated by the bounded interval [-1,1] on which values are sampled.
Only small values of parameters are sampled, therefore the resulting solutions are implicitely regularized by the scale of parameters.
In other words, the results shown in this paper can be easily explained by noting that smaller values of parameters tend to generalize better, an observation that is well known.
(THIS POINT WAS FULLY ADDRESSED DURING REBUTTAL)

- Section 5 is trivial.
The quadratic problem does not contribute in testing the hypothesis of this paper, it is constructed artificially to have larger volume for linear models, but we cannot say anything about whether other, more interesting problems display similar phenomena.
The slab dataset is slightly more interesting but it is not explained how to compute the volume of solutions in that case.

Minor:
- Linear models usually have unique solutions, however table 1 suggests that multiple solutions can be found with different train loss.
How can that be?


**Summary Of The Paper:**

This paper considers the hypothesis that the good empirical generalization of overparameterized models (neural networks in particular) may not be primarily due to an implicit bias of the optimizer (e.g. SGD), as proposed previously.
Instead, it argues that the volume in parameter space of good solutions (generalizing well) is much larger than the volume of bad solutions (generalizing poorly).
The hypothesis is tested by using three optimization algorithms: "Guess and Check", "Patern Search", "Random Greedy Search".
It is shown that those algorithms do not perform significantly worse than SGD on some toy datasets, MNIST and CIFAR.



**Summary Of The Review:**

The hypothesis stated in this paper is very interesting and deserves attention.
However, this paper falls short in properly testing it.

---

### Official Review · Reviewer_qbak · 2022-10-25

**Confidence:** 4
**Correctness:** 4
**Technical Novelty And Significance:** 2
**Empirical Novelty And Significance:** 3
**Recommendation:** 8

**Clarity, Quality, Novelty And Reproducibility:**

- I haven’t attempted to reproduce the results of this paper but the experiment descriptions are sufficiently clear that I am reasonably confident I could replicate them.
- The paper is clearly written.
- See my comments on novelty above (Weaknesses: 1.)

**Strength And Weaknesses:**

I thoroughly enjoyed reading this paper. It is well-written, the hypothesis and evaluation methods are clear, and it questions a widely-held belief that had previously not been rigorously tested (at least, not to my knowledge).

I think the principal strength in the paper comes from its evaluation of zeroth-order optimizers, both in terms of the decision boundary they induce and in terms of the generalization performance of the solutions found by these optimizers. The set of algorithms considered provide a spectrum between brute force search and gradient descent, meaning that if there had been a bias induced by gradient descent in these settings we might have expected to see a dose-response effect wherein the more gradient-like zeroth-order optimizers might exhibit better generalization or a stronger inductive bias than the guess-and-check approach.

Given the computational complexity of training larger networks with zeroth-order optimizers, I thought the paper did a good job of constructing a set of evaluation benchmarks of varying complexity in both the network and dataset size. For example, the paper evaluates the (relatively, compared to the 20-hidden unit models of section 3.2) large LeNet architecture on a significantly reduced MNIST task in order to be able to evaluate the guess and check algorithm on a high-dimensional parameter set.


The paper has two main weaknesses that I would like to see addressed.

1. The story told by the related work section is somewhat over-simplified. While much of the deep learning literature focuses on inductive bias of gradient descent and its influence on the flatness of minima, the Bayesian machine learning literature has historically taken the perspective that flat minima are good _because_ they correspond to a larger volume of parameter space. See, for example, the work of Smith et al. [1] which illustrates the connection between the existence of flat minima and the Bayesian model evidence. Further, prior works [2, 3] have studied the bias of the mapping between parameters and functions in neural networks in order to explain generalization. This somewhat reduces the novelty of the perspective taken by this paper, though I still think the analysis and experiments provided here are compelling in their own right. I recommend the authors include this discussion in the related work section to accurately position the contribution of this paper.

2. The main limitation of this work is that, understandably due to the computational complexity of zeroth-order optimization, it is not able to study the scaling behaviour of neural networks. As a result it is difficult to be confident that the findings on the bias towards simple functions observed in the small networks studied in this paper is replicated in the larger networks seen in practice. For example, it might be the case that larger networks exhibit a greater volume of poorly-generalizing minima, but that the inductive bias of SGD in these networks is enough to avoid finding them. I would be interested in seeing, potentially in the LeNet architecture on the small subset of MNIST, whether scaling the width improves or reduces generalization performance. It would be particularly interesting to me to see whether Guess & Check exhibits a similar double descent phenomenon as SGD


[1] “A Bayesian Perspective on Generalization and Stochastic Gradient Descent” Smith et al., ICLR 2018.

[2] “On the Spectral Bias of Neural Networks.” Rahaman et al., NeurIPS 2019.

[3] “Deep learning generalizes because the parameter-function map is biased towards simple functions” Guillermo Valle-Perez, et al., ICLR 2019.

**Summary Of The Paper:**

This paper challenges the folk wisdom shared by many works in the deep learning literature that neural networks generalize well due to the bias induced by the optimization algorithms we use to train them. It argues instead that the mapping between parameters and functions in neural networks is biased such that the set of solutions which generalize well has greater volume than the set of those which generalize poorly. The paper evaluates this hypothesis by comparing the performance and qualitative simplicity of solutions found by gradient descent with a number of zeroth-order optimizers, finding minimal difference between the two sets of solutions.

**Summary Of The Review:**

This paper presents an intriguing investigation into whether the good generalization properties of neural networks can be attributed solely to the implicit bias of SGD, or whether it might come from the bias of the network architecture itself towards simple functions. The experiments are clearly described and provide surprising and compelling results. While the paper could position itself better relative to prior work which also discusses the bias of the parameter-function map of neural networks, the contribution presented here is still of interest in its own right.

---

### Public Comment · ~Yimeng_Zhang2 · 2023-02-21
**Code Release Plan**

Dear Authors,

Thank you so much for such inspiring work!

Our group currently works in a related direction, and we plan to choose the guess-and-check optimizer and the zeroth-order Pattern Search optimizer as the strong baselines.

May I know your plan for sharing the code? If the code can be released in the near future, we will really appreciate it!

Waiting for your reply. Thank you so much!

Best regards,
Yimeng

---

> ### Author Response · Authors · 2023-02-21
> **It will be released soon**
>
> Hello Yimeng, Thanks for checking out our work! I am actively working on cleaning the code. I will make sure that it is released over the next week.

---

> > ### Public Comment · ~Yimeng_Zhang2 · 2023-02-21
> > **Many thanks!**
> >
> > Thanks a lot! It is really helpful!

---

> > > ### Author Response · Authors · 2023-02-28
> > > **Updated github link**
> > >
> > > Hello, I have now included link with the repository. Please let me know if you have any questions.

---

### Public Comment · ~David_Krueger1 · 2023-03-10
**Renaming existing optimizers**

Very interesting work.

I just want to note that, per Wikipedia:
* "Pattern Search" is Hill Climbing: https://en.wikipedia.org/wiki/Hill_climbing
* "Random Greedy Search" is Random Optimization: https://en.wikipedia.org/wiki/Random_optimization

---

> ### Author Response · Authors · 2023-03-13
> **Thanks for the note!**
>
> Hello David, thanks for your feedback!  We would like to point out that what we call 'Pattern Search' is exactly the algorithm depicted on the Wikipedia page for Pattern Search https://en.wikipedia.org/wiki/Pattern_search_(optimization), but we agree with you that this is similar to Hill Climbing.  We chose a descriptive name for Random Greedy Search, and we agree that random optimizers go by other names too.

---

### Decision · Program_Chairs · 2023-01-20

**Decision:**

Accept: notable-top-25%

**Justification For Why Not Higher Score:**

It could also be an oral given how thought provoking the paper is. On the other hand, the study of mainly low sample regime puts a caveat on the generalizability of their observation on more realistic tasks (this was partially addressed in the revision though).

**Justification For Why Not Lower Score:**

This is one of the rare papers which really makes us rethink an important assumption that was previously made in the literature. It is more interesting than other, perhaps more polished papers, but that  are more incremental in what they mean.

**Metareview: Summary, Strengths And Weaknesses:**

Summary: This paper challenges the folk wisdom shared by many works in the deep learning literature that neural networks generalize well due to the bias induced by the optimization algorithms we use to train them. It argues instead that the mapping between parameters and functions in neural networks is biased such that the set of solutions which generalize well has greater volume than the set of those which generalize poorly. The paper evaluates this hypothesis by comparing the performance and qualitative simplicity of solutions found by gradient descent with a number of zeroth-order optimizers, finding minimal difference between the two sets of solutions.

All three reviewers thoroughly enjoyed this paper. They expressed some initial concerns in their reviews that have been well-addressed by the authors in a revision and they all have updated their score accordingly. Reviewer Qs6Q questioned the pertinence of the "Pattern Search" and "Random Greedy Search" in the story for this paper, and instead recommended the authors to focus exclusively on the "Guess and Check" method -- this explains their final score of 5. The other two reviewers, on the other hand, appreciated the overall story, and recommend acceptance as is.

While this AC is sympathetic to the points brought by Qs6Q about the story rewriting, they think that this paper already presents a very interesting and thought provoking investigation that is worthy of being showcased at ICLR, thus recommending a spotlight.

== Other comments:

The authors should also mention in the related work some past papers that studied the ability of neural networks to perform surprisingly well with just random initialization (no training). See for example "Weight Agnostic Neural Networks" by Gaier & Ha, NeurIPS 2019, and their references therein. "Guess and Check" is taking this in another direction (less tuned random initialization; but instead allow multiple guesses to be checked).

**Note From Pc:**

if the above contains the word "oral" or "spotlight" please see: "oral" presentation means -> notable-top-5% and "spotlight" means -> notable-top-25%. As stated in our emails, we are disassociating presentation type from AC recommendations